# [^68^Ga]DOTATOC PET/CT Radiomics to Predict the Response in GEP-NETs Undergoing [^177^Lu]DOTATOC PRRT: The “Theragnomics” Concept

**DOI:** 10.3390/cancers14040984

**Published:** 2022-02-16

**Authors:** Riccardo Laudicella, Albert Comelli, Virginia Liberini, Antonio Vento, Alessandro Stefano, Alessandro Spataro, Ludovica Crocè, Sara Baldari, Michelangelo Bambaci, Desiree Deandreis, Demetrio Arico’, Massimo Ippolito, Michele Gaeta, Pierpaolo Alongi, Fabio Minutoli, Irene A. Burger, Sergio Baldari

**Affiliations:** 1Nuclear Medicine Unit, Department of Biomedical and Dental Sciences and Morpho-Functional Imaging, University of Messina, 98125 Messina, Italy; antvento@hotmail.it (A.V.); Alessandro.spataro@outlook.it (A.S.); croceludovica@gmail.com (L.C.); fminutoli@unime.it (F.M.); sbaldari@unime.it (S.B.); 2Ri.MED Foundation, 90134 Palermo, Italy; acomelli@fondazionerimed.com; 3Department of Nuclear Medicine, University Hospital Zürich, University of Zürich, 8091 Zürich, Switzerland; Irene.burger@usz.ch; 4Nuclear Medicine Unit, Fondazione Istituto G.Giglio, 90015 Cefalù, Italy; alongi.pierpaolo@gmail.com; 5Nuclear Medicine Unit, Department of Medical Sciences, University of Turin, 10126 Turin, Italy; v.liberini@gmail.com (V.L.); desiree.deandreis@unito.it (D.D.); 6Nuclear Medicine Department, S. Croce e Carle Hospital, 12100 Cuneo, Italy; 7Institute of Molecular Bioimaging and Physiology, National Research Council (IBFM-CNR), 90015 Cefalù, Italy; Alessandro.stefano@ibfm.cnr.it; 8Nuclear Medicine Department, Cannizzaro Hospital, 95126 Catania, Italy; sarabaldari@yahoo.it (S.B.); ippolitomas@yahoo.it (M.I.); 9Department of Nuclear Medicine, Humanitas Oncological Centre of Catania, 95125 Catania, Italy; Michelangelo.bambaci@ccocatania.it (M.B.); Demetrio.arico@ccocatania.it (D.A.); 10Section of Radiological Sciences, Department of Biomedical Sciences and Morphological and Functional Imaging, University of Messina, 98125 Messina, Italy; mgaeta@unime.it; 11Department of Nuclear Medicine, Kantonsspital Baden, 5404 Baden, Switzerland

**Keywords:** ^177^Lu, artificial intelligence, [^68^Ga]DOTATOC PET, GEP NET, machine-learning, PRRT, delta radiomics

## Abstract

**Simple Summary:**

The radiological response assessment of neuroendocrine tumors (NET) to peptide receptor radionuclide therapy (PRRT) using [^177^Lu]DOTATOC is still suboptimal due to the high variability in targeted somatostatin receptor 2 (SSTR-2) expression and histological heterogeneity among patients with well-differentiated NET. Promising and innovative laboratory assays have been proposed, but they are highly costly and not easily accessible. Machine learning offers new opportunities to provide quantitative characteristics from molecular images that cannot be appreciated by the human eye. We therefore retrospectively analyzed [^68^Ga]DOTATOC PET/CT images before and after complete [^177^Lu]DOTATOC PRRT in well-differentiated progressive, metastatic gastroenteropancreatic NET and obtained radiomics features as new and reliable imaging parameters that correlate to the response to PRRT and might be used for improved patient selection in the future.

**Abstract:**

Despite impressive results, almost 30% of NET do not respond to PRRT and no well-established criteria are suitable to predict response. Therefore, we assessed the predictive value of radiomics [^68^Ga]DOTATOC PET/CT images pre-PRRT in metastatic GEP NET. We retrospectively analyzed the predictive value of radiomics in 324 SSTR-2-positive lesions from 38 metastatic GEP-NET patients (nine G1, 27 G2, and two G3) who underwent restaging [^68^Ga]DOTATOC PET/CT before complete PRRT with [^177^Lu]DOTATOC. Clinical, laboratory, and radiological follow-up data were collected for at least six months after the last cycle. Through LifeX, we extracted 65 PET features for each lesion. Grading, PRRT number of cycles, and cumulative activity, pre- and post-PRRT CgA values were also considered as additional clinical features. [^68^Ga]DOTATOC PET/CT follow-up with the same scanner for each patient determined the disease status (progression vs. response in terms of stability/reduction/disappearance) for each lesion. All features (PET and clinical) were also correlated with follow-up data in a per-site analysis (liver, lymph nodes, and bone), and for features significantly associated with response, the Δradiomics for each lesion was assessed on follow-up [^68^Ga]DOTATOC PET/CT performed until nine months post-PRRT. A statistical system based on the point-biserial correlation and logistic regression analysis was used for the reduction and selection of the features. Discriminant analysis was used, instead, to obtain the predictive model using the k-fold strategy to split data into training and validation sets. From the reduction and selection process, HISTO_Skewness and HISTO_Kurtosis were able to predict response with an area under the receiver operating characteristics curve (AUC ROC), sensitivity, and specificity of 0.745, 80.6%, 67.2% and 0.722, 61.2%, 75.9%, respectively. Moreover, a combination of three features (HISTO_Skewness; HISTO_Kurtosis, and Grading) did not improve the AUC significantly with 0.744. SUV_max_, however, could not predict the response to PRRT (*p* = 0.49, AUC 0.523). The presented preliminary “theragnomics” model proved to be superior to conventional quantitative parameters to predict the response of GEP-NET lesions in patients treated with complete [^177^Lu]DOTATOC PRRT, regardless of the lesion site.

## 1. Introduction

Neuroendocrine tumors (NETs) are heterogeneous and rare neoplasms, even if their incidence rate has increased consistently during the last decades. Gastroenteropancreatic (GEP) NETs represent the most common subtype, covering up to 70% of all NET. Functional GEP NET release hormones that can cause symptoms, being consequently diagnosed earlier, while non-functional GEP NETs are more frequent and usually diagnosed at an advanced stage [1,2]. According to the recent 2019 WHO classification, GEP NET can be divided into G1, G2, G3, small-cell type (SCNEC), large-cell type (LCNEC), and mixed NET (MiNET) based on differentiation, grade, mitotic rate, and ki-67 index [3]. Well-differentiated GEP NET (G1-G2-G3, or MiNET in case that the prevalent component is well-differentiated) are mostly characterized by slow growth and good survival, even in the presence of synchronous liver metastases at diagnosis. Otherwise, poorly differentiated GEP neuroendocrine carcinomas (SCNEC, LCNEC, or MiNET in case that the prevalent component is poorly differentiated) are more aggressive with worse prognosis [4,5]. A precise assessment of primary (T) and eventual widespread disease (N and M) is of the utmost importance to select the best therapeutic approach for GEP NET. In this scenario, targeted somatostatin receptor 2 (SSTR 2) molecular imaging with positron emission tomography (PET)/computed tomography (CT) or magnetic resonance imaging (MRI) plays a significant role. Therapeutic options for GEP NET include curative surgery when feasible, interventional radiology, somatostatin analogues, interferon, chemotherapy, targeted drugs (i.e., everolimus, sunitinib), selective internal radiotherapy, and peptide receptor radionuclide therapy (PRRT) using radiolabelled somatostatin analogues [6]. PRRT represents an effective treatment for metastatic or inoperable NET, recently approved in Europe, USA, and Canada for GEP forms [7]. PRRT is included in the theragnostics scenario, enabling, through a unique radiopharmaceutical administration for multiple cycles, a molecularly targeted therapeutic procedure (i.e., beta minus emission of ^177^Lu) and biodistribution imaging (i.e., gamma emission of ^177^Lu). However, although PRRT is effective in the majority of cases, approximately 15–30% of patients will progress during PRRT and can benefit from timely adjustments, therapy combinations, rapid sequencing, or alternatives. Furthermore, the Delphic consensus for GEP NET response to therapy assessment defined both the RECIST 1.1 criteria and PET parameters as suboptimal due to the high variability in SSTR expression, the different histological patterns related to disease heterogeneity, heterogeneous responses, and lack of standardized criteria for molecular imaging [8]. In addition, biochemical assessment of tumor markers, such as Chromogranin A is also suboptimal [9]. However, promising and innovative approaches, such as NET TEST, have been proposed but they are highly costly and not easily accessible [10]. Therefore, the identification of new and reliable quantitative imaging parameters could be crucial to better address eligible candidates and to assess the response to PRRT, early selecting the best therapeutic opportunity, avoiding high-costly treatments [11] and related toxicities. In this scenario, radiomics is a promising technique based on advanced mathematics and statistics that aims to provide quantitative characteristics (features) from biomedical images of diverse nature that cannot be assessed by the human eye [12]. In other words, radiomics assumes that every part of the image (even the smallest) may include tumor features that may be potentially related to patient outcomes [13,14], response to therapy [15], and molecular profile [16], supporting medical decisions. Features of several orders can be extracted and each of those may be related to a precise meaning as the I order features containing information on shape and statistics deriving from the histogram describing the distribution of grey values in the selected lesion, or the II or higher orders encompassing information about the relationships between adjacent voxels. A few studies already assessed the potential application of machine-learning (ML) in GEP-NET to predict response to PRRT. However, such studies referred to very limited populations [17,18], heterogeneous cohorts, or considered only predefined features [18,19,20,21]. Therefore, we aimed to develop a more robust radiomics (“radiOMICS”) predictive model of response analyzing [^68^Ga]DOTATOC PET/CT images before and after complete [^177^Lu]DOTATOC PRRT (“THERAGNOstics”) in well-differentiated, progressive, metastatic GEP NET, namely “Theragnomics” that can be applied in a clinical decision support system (CDSS).

## 2. Materials and Methods

### 2.1. Patients

In this retrospective study, we included all consecutive well-differentiated GEP NET patients who, between 1 April 2013 and 30 November 2019, underwent a baseline [^68^Ga]DOTATOC PET/CT within 2 months before beginning the PRRT with [^177^Lu]DOTATOC, and a follow-up [^68^Ga]DOTATOC PET/CT available within 9 months after the last PRRT cycle. Chromogranin A (CgA) was also assessed before each PRRT cycle and at the end of the treatment. Clinical, laboratory, and [^68^Ga]DOTATOC PET/CT follow-up data were collected for a period of at least 3 months after the last cycle. Patients were not eligible if: (a) they were under 18 years of age; (b) lack of follow-up/baseline imaging and clinical data; (c) patients with other concomitant oncological pathology. In Figure 1, we described the study workflow. The study was approved by the institutional review board (668-18/20), conducted according to the Declaration of Helsinki principles and good clinical practice guidelines, and written informed consent specifying the potential use of anonymized data for research purposes was obtained for each patient.

### 2.2. [^68^Ga]DOTATOC PET/CT

All GEP NET patients underwent [^68^Ga]DOTATOC PET/CT with the same scanner at different institutions (GE Discovery ST, Discovery ST, and Discovery 690; Siemens Biograph Horizon; Philips Gemini GXL 16) before and after complete PRRT for staging and restaging purposes following the current guidelines [22]. [^68^Ga]DOTATOC PET/CT images were acquired 60 min after an administered [^68^Ga]DOTATOC dose of 2 MBq/kg and co-registered with low-dose CT. The PET scans were validated for proper quantification and quality.

### 2.3. Image Analysis

Following the current guidelines, [^68^Ga]DOTATOC PET/CT positivity was confirmed in the case of non-physiologically uptake or higher uptake than background activity. In comparison with baseline, [^68^Ga]DOTATOC PET/CT follow-up after PRRT determined the status of response to therapy for each lesion in terms of disease progression (PD, increase in lesion size/SUV_max_ of at least 25%) vs. stability (SD, increase-reduction in lesion size/SUV_max_ < 25%), reduction (PR, decrease in lesion size/SUV_max_ of at least 25%), or disappearance (CR) [21]. All PET/CT images were qualitatively analyzed with a dedicated workstation and were interpreted by D.A. and M.I. (nuclear medicine physicians with 16 and 20 years of experience, respectively).

### 2.4. PRRT

All patients completed full PRRT (at least 5 cycles) that began within 2 months after baseline [^68^Ga]DOTATOC PET/CT. PRRT was performed only in patients with haemoglobin ≥ 8 g/dL, white blood cells ≥ 3000/mmc, platelets ≥ 75,000/mmc, creatinine ≤ 1.70 mg/dL, and creatinine clearance ≥ 40 mL/min, according to published guidelines [23]. Preparation of [^177^Lu]DOTATOC was carried out following established procedures [24]. Therapy response was routinely assessed on an individual lesion level.

### 2.5. Radiomics [^68^Ga]DOTATOC PET/CT Analysis

LifeX [25] is an analysis software compliant with the Image Biomarker Standardization Initiative (IBSI) [26] that allows the automatic extraction of radiomics features from biomedical images. For each patient, all [^68^Ga]DOTATOC-positive lesions that were clearly discriminated, non-confluent, and of minimal size of 16 voxels were selected. PET images were imported to LifeX and a 2D-circular region of interest (ROI) were drawn around every lesion. ROIs had a minimum size of 0.443 cm^3^ (corresponding to at least 16 voxels) to allow for a consistent textural feature calculation. ROI size was adjusted to the size of the lesions, without incorporating adjacent tissue; ROI size was adjusted to the size of the lesions, without incorporating adjacent tissue. In this way, using an absolute intensity rescaling factor of 0–60 of the SUV (64 bins, 0.95 fixed bin width), 65 radiomics features were automatically extracted for each lesion. In addition, five clinical features were also considered: grading (G1-G2-G3), number of PRRT cycles, PRRT cumulative activity, pre- and post-PRRT CgA values. All the features (imaging and clinical) were correlated with the response data. Specifically, due to the redundancy, heterogeneity, and uncertainty of the information represented by the radiomics features, we used an innovative mixed descriptive-inferential sequential approach [27,28] for the feature selection and reduction process. For each feature, the point biserial correlation (pbc) index between features and the dichotomic outcome (PD vs. SD, PR, CR) was calculated, sorting the features in pbc descending order. Then, a cycle started to add one column at a time, performing a logistic regression analysis by comparing the *p*-value of each iteration and stopping in the case of a growing *p*-value. Accordingly, the features with valuable association with the outcome were identified and assessed (singularly and in combination) for response to PRRT prediction. Finally, the discriminant analysis (DA) was used for implementing the classification model using the k-fold strategy to split data into training and validation sets. In this way, the PET studies were divided into k-folds. One of the folds was used as the validation set and the remaining folds were combined in the training set. This process was repeated k-times using each fold as the validation set and the other remaining sets as the training set. In our study, k = 5 was empirically determined by trial-and-error strategy (k range: 5–15, step size of 5). To ensure disjointed validation sets, the leave-one-out approach was not adopted. In this way, more robust results can be obtained in implementing the classification model [29]. For the most significant features, we also assessed the percentage difference value before (T0) and after PRRT (T1) in terms of delta radiomics, translating the pre-PRRT [^68^Ga]DOTA-peptide PET/CT ROI in the same lesion area of the follow-up performed within nine months after PRRT. The delta radiomics was then calculated using the following formula:Δ = 100 ∗ (Feature T1 − Feature T0)/Feature T0

Finally, we performed a per-district analysis (lymph node, liver, and bone) evaluating all the pre-PRRT PET/CT features in response to PRRT prediction. In Figure 2, we described the radiomics workflow.

### 2.6. Statistical Analysis

Quantitative variables were expressed as mean ± standard deviation. Descriptive analyses were used to display patient data as mean and range. The t-test was used to compare means. The differences of the most significant features and delta radiomics between responders and non-responders were compared using a non-parametric Mann–Whitney U test. The ability of the most significant radiomics features to predict the response to PRRT was assessed with receiver operating characteristics (ROC) analysis, and the Youden index was used for the maximization of specificity and sensitivity. The area under the curves (AUC) was reported. In addition, a site-dependent sub-analysis was performed for the most represented districts of our cohort (lymph node, liver, and bone), evaluating both the pre-PRRT PET/CT parameters, radiomics features, and the delta radiomics for the most significant parameters in the response to PRRT prediction.

Statistical analyses were performed by R.L. and V.L. (nuclear medicine physicians), A.C. and A.St. (computer science PhDs). Statistical analyses were performed using SPSS statistics software, version 26 (IBM, Armonk, NY, USA), and a *p*-value of less than 0.05 was considered statistically significant.

## 3. Results

A total of 38 GEP NET patients with a median age of 58 years (range 35–79; mean 59.4 ± 10.3 y; 15 out of 38 female) were retrospectively included and underwent a baseline [^68^Ga]DOTATOC PET/CT (mean activity 151.1 ± 55.5 MBq, range 93–330 MBq; median 120.5 MBq) a mean of 1.4 ± 0.7 months (0–2) before complete PRRT with a median cumulative dose of 29.0 GBq (23.9–32.8 GBq), followed by [^68^Ga]DOTATOC PET/CT (mean activity 165 ± 62.6 MBq, range 93–330 MBq; median 128.5 MBq) a mean of 8.7 ± 1.1 months (3–9) after the last PRRT cycle. The primary sites originated from the pancreas in 17 out of 38, ileum 14 out of 38, colon three out of 38, stomach two out of 38, and jejunum two out of 38. Grading was distributed as follows: 9/38 G1, 27/38 G2, 2/38 G3. [^177^Lu]DOTATOC PRRT was performed a median of five cycles (5–7; total 200; mean 5.3 ± 0.5) with a mean administered activity of 29 ± 1.5 GBq. [^177^Lu]DOTATOC labelling yield >99% was reached in all cases according to guidelines [23]. Baseline CgA was 277 ng/mL (17–1315; mean 394.7 ± 376.1 ng/mL), while follow-up CgA was 125.5 ng/mL (16–1630; mean 380.5 ± 426 ng/mL). Patients’ characteristics and scanner types are summarized in Table 1. The patients’ inclusion diagram is reported in Appendix A.

### 3.1. [^68^Ga]DOTATOC PET/CT Findings

At baseline [^68^Ga]DOTATOC PET/CT, we obtained 324 SSTR-positive lesions with at least 16 voxels. Based on their location, lesions were divided as follows: 169 in 324 liver, 91 in 324 lymph nodal, 42 in 324 bone lesions, and 22 in 324 parenchymal (different than liver). At the qualitative assessment of follow-up [^68^Ga]DOTATOC PET/CT, 133 in 324 lesions were classified as PD and 191 lesions as responsive to therapy (SD + PR + CR, Table 1).

### 3.2. Radiomics Analysis

Through LifeX software, 65 features were extracted from baseline [^68^Ga]DOTATOC PET/CT for each lesion. All features (65 from PET and five from clinical data) were then correlated with the response data at follow-up. The complete list of extracted features is provided in Appendix A. From the reduction and selection process, the combination of three features, two from PET (HISTO_Skewness; HISTO_Kurtosis) and one clinical (Grading) proved able to predict each lesion’s response to PRRT in terms of progression vs. positive results, regardless of their nature (parenchymal, lymph nodes, bone lesions), with an AUC ROC, sensitivity, and specificity of 0.744, 66.4%, and 70.3%, respectively. However, the best predictive result was obtained for HISTO_Skewness, with an optimal cut-off at 2.45 reaching an AUC ROC, sensitivity, and specificity of 0.745, 80.6%, and 67.2%, respectively. Moreover, HISTO_Kurtosis, with an optimal cut-off at 6.94 reached an AUC ROC, sensitivity, and specificity of 0.722, 61.2%, and 75.9%, respectively. Differently, the SUV_max_ was not significant (*p* = 0.49) to predict the response to PRRT in terms of progression vs. objective benefit or response (AUC ROC 0.523, sensitivity 36.7%, specificity 63.3%), as shown in Figure 3.

Furthermore, the two aforementioned features (HISTO_Skewness and HISTO_Kurtosis) were significantly higher (*p* < 0.001) in non-responders’ lesions than in responders’ lesions before and after PRRT, as shown in Appendix A. Indeed, for such features, we also assessed the delta radiomics, as described in Section 2. After PRRT, in responsive lesions (SD + PR + CR) we observed a mean percentage reduction for ΔHISTO_Skewness (−3.31% ± 664.3%) and a mean percentage increase for ΔHISTO_Kurtosis (15.98% ± 71.4%). Differently, for progressive lesions (PD), we observed a higher mean percentage increase for ΔHISTO_Skewness (112.54% ± 348.3%; *p* = 0.209) and for ΔHISTO_Kurtosis (5.81% ± 52.3%), less evident than responsive/stable lesions (*p* = 0.255).

### 3.3. Lesions’ Per-Site Sub-Analysis

We performed a site-dependent sub-analysis for the most represented districts of our cohort (lymph node, liver, and bone), evaluating all the most significant pre-PRRT PET/CT features in response to PRRT prediction, also considering the **Δ**HISTO_Skewness and **Δ**HISTO_Kurtosis.

The following PET features showed a statistically significant difference between responder and non-responder lesions at the Mann–Whitney test: for the lymph node lesions (n = 91; 41 of 91 non-responsive and 50 of 91 responsive), SUV_min_ and SUV_mean_ (both *p* < 0.028); metabolic tumor volume (MTV; *p* < 0.0028); HISTO_Skewness and HISTO_Kurtosis (both *p* < 0.028); shape (mL, *p* = 0.012). For liver lesions (n = 169; 61 of 169 non-responsive and 108/169 responsive), MTV (*p* < 0.001), all HISTO features (*p* < 0.041), GLCM_Energy (*p* = 0.05), and GLCM_Entropy (*p* = 0.048). Finally, for bone lesions (n = 42; 24 of 42 non-responsive and 18 of 42 responsive), only HISTO_Skewness and HISTO_Kurtosis (both *p* < 0.014) showed a statistically significant difference between responder and non-responder lesions. Moreover, in the sub-analysis for districts, the SUV_max_ was not significant in predicting response to PRRT in terms of progression vs. objective benefit or response (*p* > 0.05), with the only exception for the bone district (*p* = 0.047). The mean values of HISTO_Skewness, HISTO_Kurtosis, and SUV_max_ for responder and non-responder patients in the three districts are presented in Table 2.

For HISTO_Skewness and HISTO_Kurtosis, optimal cut-offs for predicting PRRT responder vs. non-responder lesions were defined using the ROC curve, as shown in Appendix A. For the lymph node district, the AUC of HISTO_Skewness was 0.67 (best cut-off at 2.45 with a sensibility and specificity of 76% and 60%, respectively), while the AUC of HISTO_Kurtosis was 0.64 (best cut-off at 8.10 with a sensibility and specificity of 76% and 58%, respectively). For the liver district, the AUC of HISTO_Skewness was 0.76 (best cut-off at 1.94 with a sensibility and specificity of 87% and 67%, respectively), while the AUC of HISTO_Kurtosis was 0.75 (best cut-off at 6.55 with a sensibility and specificity of 87% and 68%, respectively). For the bone district, the AUC of HISTO_Skewness was 0.73 (best cut-off at 3.33 with a sensibility and specificity of 79% and 78%, respectively), while the AUC of HISTO_Kurtosis was 0.72 (best cut-off at 15.33 with a sensibility and specificity of 79% and 78%, respectively). For the other aforementioned parameters, the ROC curve was not informative (AUC < 0.5).

Finally, in Table 3, we summarized the results of the per-site sub-analysis performed on the Δradiomics of HISTO_Skewness and HISTO_Kurtosis. Accordingly, only ΔHISTO_Skewness for the liver district and ΔHISTO_Kurtosis for the bone district showed a statistically significant difference between PRRT responder and non-responder lesions (*p* = 0.031 and *p* = 0.022, respectively). However, the ROC curve for these two parameters was not informative (AUC < 0.6), probably related to the small sample size.

## 4. Discussion

It is well known that the heterogeneity of GEP-NET limits the tumor grading based on biopsy samples [30]. Furthermore, response to PRRT assessment is still suboptimal using either, conventional imaging or tumor markers [8]. Considering the impact in terms of side-effects and the high cost of advanced therapy for progressive, metastatic, or inoperable GEP NET [11], the identification of new, reproducible, and easily accessible biomarkers seems to be crucial towards a non-invasive and reliable whole-body assessment of tumor grading. Due to its ability to visualize whole-body tumor burden on a molecular level, PET-based tumor heterogeneity improved the intra-individual assessment of tumor biology. In our model, the [^68^Ga]DOTATOC PET/CT radiomics features “HISTO_Skewness” and “HISTO_Kurtosis” were able to predict the PRRT response based on a lesion for primary tumors as well as metastasis regardless of the origin with an AUC ROC, sensitivity, and specificity of 0.745, 80.6%, 67.2%, and 0.722, 61.2%, 15.9%, respectively, vs. 0.523 for the SUV_max_ that was not significant to predict the response to PRRT (*p* = 0.49). Moreover, the combination of two radiomics features (HISTO_Skewness; HISTO_Kurtosis) together with one clinical feature (Grading) was able to predict the PRRT response with an AUC ROC, sensitivity, and specificity of 0.744, 66.4%, and 70.3%, respectively, but did not improve the accuracy over the HISTO_Skewness.

So far, very few studies have investigated the role of ML in the prediction of response to PRRT in GEP NET patients. Wetz et al. have reported on the predictive role of “asphericity” in GEP-NET patients enrolled for PRRT [31]. They observed that a higher level of “asphericity” was associated with poorer outcomes. However, compared to our study investigating [^68^Ga]DOTATOC PET/CT, features were derived from [^111^In]DTPA0-octreotide scintigraphy, which has a lower affinity to SSTR2 compared to PET radiopharmaceuticals, and different image modalities than PET/CT and/or PET/MRI. More recently, Önner et al. assessed the value of two predefined first-order features, “skewness” and “kurtosis” (interestingly the same to our study, that we obtained in a non-predefined way as described in the material and methods section), in the prediction of response to PRRT in 22 GEP NET patients for a total of 326 lesions [21]. Differently from our study, they considered SD as a non-response to PRRT, even if in the clinical practice the stability of disease is a warranted result in this scenario considering that PRRT is approved for progressive, metastatic, and usually heavily treated NET patients. Similar to our results, they observed that such features were significantly higher in non-responder patients (*p* < 0.001 for skewness and *p* = 0.004 for kurtosis, vs. a *p* < 0.001 in our study for both). Moreover, they assessed the features’ predictive power in PRRT response assessment for skewness and kurtosis singularly (without any clinical/biochemical parameters), reaching less significant results compared to our study (AUC ROC of 0.619 for skewness and 0.518 for kurtosis vs. 0.745 and 0.722 in our paper, respectively; cut-offs 2.45 and 6.94). In a different scenario from our paper (survival analysis), Werner et al. described their experience in a multicentric cohort of 142 NET patients (108/142 GEP NET) applying predefined features. The authors reported that four features, namely “entropy” (similar to our results for lymph node lesions), “correlation”, “short-zone emphasis”, and “homogeneity”, provided a significant distinction between responders from non-responders. Furthermore, “entropy” proved to be independently associated with progression-free survival (PFS) and overall survival (OS) while “skewness” was independently associated with OS. Moreover, conventional PET parameters did not predict any of these outcomes [19]. Similarly, in our study, we observed that the SUV_max_ was not significant to predict the response to PRRT (*p* = 0.49), and only slightly significant for the distinction between PRRT responder and non-responder bone lesions (*p* = 0.047). The same group later assessed (with predefined features) 31 G1-G2 pancreatic NET patients who underwent [^177^Lu]DOTATATE PRRT. They observed that differently from conventional PET parameters, a cut-off > 6.7 for “entropy” reached a significant predictive ability for longer OS (AUC 0.71) [20]. Moreover, in our study, for the most statistically significant PET features, we assessed the percentage variations in terms of delta radiomics: in responsive/stable lesions, we observed a mean % reduction for ΔHISTO_Skewness (−3.3% ± 664.3%) and a mean % increase for ΔHISTO_Kurtosis (16% ± 71.4%); for progressive lesions, we observed a mean % increase for ΔHISTO_Skewness (112.5% ± 348.3%) and ΔHISTO_Kurtosis (5.8% ± 52.3%), less evident than for responsive/stable lesions. In a small, heterogeneous NET cohort [18], Weber et al. applied textural analysis to [^68^Ga]DOTATOC PET/MRI liver lesions before and after PRRT at different dosages/radiopharmaceuticals using only predefined features. In terms of delta radiomics, they observed that patients undergoing therapy with somatostatin-analogue (SSA) showed a trend in “entropy” decrease (−0.07  ±  0.16) when compared to patients undergoing PRRT (0.14  ±  0.43).

In our preliminary experience, we aimed to give weight to a predictive model of response to PRRT based on the most significant [^68^Ga]DOTATOC PET/CT features. In the innovative but uncertain setup of radiomics applied to GEP-NET, we tried to reproduce a real-life scenario: well-differentiated GEP-NET who underwent [^68^Ga]DOTATOC PET/CT with different scanners before and after complete PRRT. In the lesion progression prediction, a HISTO_Skewness = 2.45 reached an AUC ROC of 0.745 (sensitivity 80.6%, specificity 67.2%) and a HISTO_Kurtosis = 6.94 reached an AUC ROC of 0.722 (sensitivity 61.2%, specificity 75.9%), with similar results if considered together with clinical parameters. Different features’ behavior needs to be further investigated: at Δradiomics analysis, we observed different lesions’ conducts according to the presence of response (a mean reduction in HISTO_Skewness and a more evident increase in HISTO_Kurtosis) or progression (a mean increase in HISTO_Skewness and, less evident, HISTO_Kurtosis) after complete PRRT. In other words, in responsive lesions after PRRT we observed a mean percentage reduction of the “asymmetry” (namely, Skewness) and a more evident increase in the “discrepancy of the considered histogram from the ordinary one” (namely, Kurtosis) than non-responsive lesions. Moreover, in the district sub-analysis, we observed that Δradiomics has a different tendency to increase or decrease for each feature, thus further reflecting NET’s heterogeneity in the liver (often extensive lesions with central necrosis) bone (often mixed and small lesions) and lymph node (possible desmoplastic reaction) [32].

As already stated, both [^68^Ga]DOTATOC PET/CT SUV_max_ and PET lesion volume are considered suboptimal parameters to assess the response to PRRT. Therefore, the potential added value of radiomics features is to provide prognostic additional information to conventional parameters, and HISTO_Skewness and HISTO_Kurtosis belong to this subset of features, as previously demonstrated [17]. The opportunity to assess for each patient the single lesion’s heterogeneity and predict each lesion’s response to PRRT would enhance physicians to early address patients to the best options of care, reducing costs and potential toxicities [11], improving quality of life and survival. The small sample size represents a limitation for the statistics of this preliminary analysis. However, we considered each patient’s lesion singularly (n = 324) and the cohort is a rather homogeneous group of well-differentiated GEP NETs who completed full [^177^Lu]DOTATOC PRRT. Furthermore, images were derived from different PET/CT scanners with potential variations in reconstruction algorithms (no harmonization was performed). Nonetheless, this study may be more similar to the real-life scenario, and the results of the testing features were also confirmed in the validation cohort, thus highlighting features’ robustness and reproducibility. In addition, radiomics features were extracted only from the most active part of [^68^Ga]DOTATOC positive lesions to construct the model, and the remaining tissue in the image may still contain invisible but useful data. To analyze the entire images, 3D deep learning methods will be necessary for the “Theragnomics” scenario. The use of deep learning algorithms might also allow us to eliminate any potential time-consuming ROI placement. The deep learning algorithm will be responsible for the entire radiomics process in a completely automatic way, from the segmentation process to the feature extraction process to the predictive model implementation, avoiding the use of LifeX or similar software, and consequently eliminating any user-dependence. Furthermore, our study focused on PET-based radiomics only without any comparison with survival. A combination with CT imaging analysis may improve the performance of the prediction model and should be evaluated in future larger studies, including more clinical/biochemical data and external validation to evaluate the possible association of PET district-based semi-quantitative parameters with the outcome.

## 5. Conclusions

The presented preliminary “theragnomics” model proved to be superior to conventional quantitative parameters to predict the response of GEP-NET lesions in patients treated with complete [^177^Lu]DOTATOC PRRT, regardless of the lesion site.

## Figures and Tables

**Figure 1 cancers-14-00984-f001:**
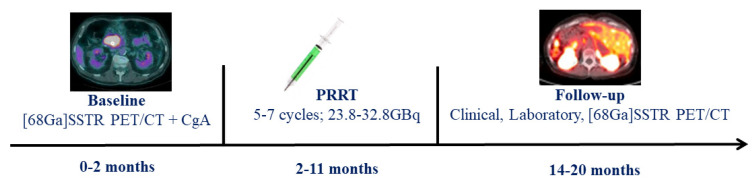
Study workflow.

**Figure 2 cancers-14-00984-f002:**
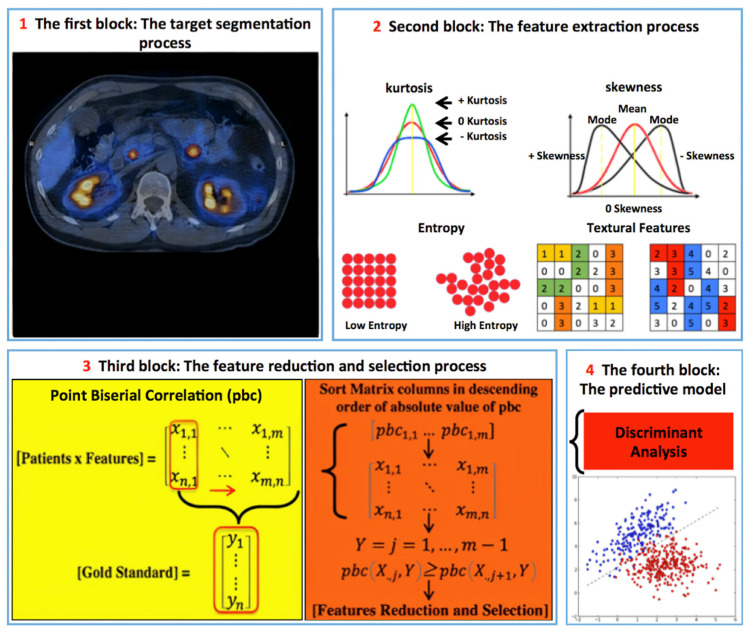
Radiomics’ workflow. (**1**) 324 GEP NET lesions with high SSTRs expression in [^68^Ga]DOTATOC PET were analyzed (LIFEx) placing a 2D-circular ROI (at least 16 voxels, 0.443 cm^3^) on the lesion’s part with the highest SUV_max_. (**2**) 65 features from each lesion (parenchyma, lymph nodes, bones) + additional features: Pre-PRRT CgA values and grading (G1-G2-G3) were assessed. (**3**) Descriptive-inferential sequential approach for feature reduction and selection; for each feature, the point biserial correlation index between the features and the dichotomic variable (0 PD vs. 1 SD, CR, PR) was calculated, sorting the features in descending order. Then, a cycle started to add one column at a time performing a logistic regression analysis comparing the *p*-value of each iteration and stopping in case of a growing *p*-value. (**4**) Discriminant analysis was then used for feature classification using the most discriminative ones identified in the previous step.

**Figure 3 cancers-14-00984-f003:**
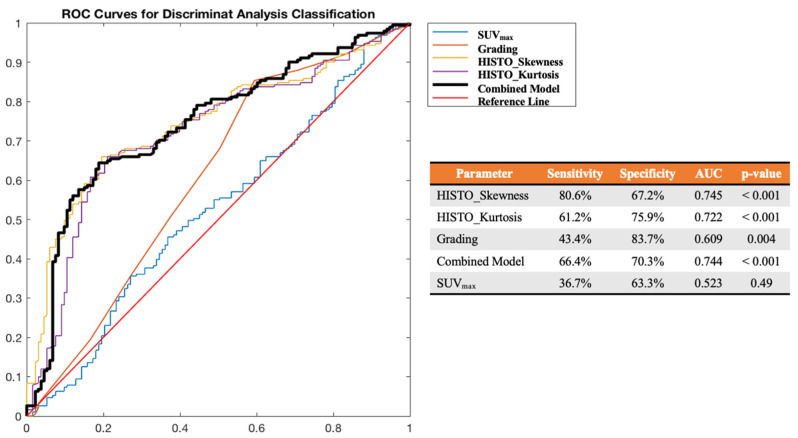
ROC curve analysis for HISTO_Skewness, HISTO_Kurtosis, Grading, their combination (Combined Model) and SUV_max_ in the prediction of response to PRRT (early FU status) in terms of PD vs. positive results (SD, PR, CR).

**Table 1 cancers-14-00984-t001:** Patients’ main characteristics.

**Patients’ Number** **(female–male)**	38 (15 F—23 M)
**Mean/median age (Range)**	59.4 ± 10.3 y/58 y (35–79)
**Mean/median administred activity (range)**	29 ± 1.5 GBq/29 GBq (23.9–32.8)
**Mean/median PRRT cycles (range)**	5.3 ± 0.5/5 (5–7)
**GEP NET origin**	
Pancreas	17/38 (45%)
Ileum	14/38 (37%)
Colon	3/38 (8%)
Stomach	2/38 (5%)
Jejunum	2/38 (5%)
**Grading (*n*)**	
G1	9/38 (23.7%)
G2	27/38 (71%)
G3	2/38 (5.3%)
**Lesions’ distribution**	
Bone Lesions	42/324 (12.9%)
Lymph nodal Lesions	91/324 (28.1%)
Liver Lesions	169/324 (52.2%)
Parenchimal Lesions (no liver)	22/324 (6.8%)
**Singular lesion response to PRRT**	
PD	133/324 (41%)
SD	79/324 (24.4%)
PR	92/324 (28.4%)
CR	20/324 (6.2%)
**Lesions’ distribution according to response (SD, PR, CR) and grading**	
G1	28/82 (34.1%)
G2	157/232 (67.7%)
G3	6/10 (60%)
**Scanner types**	**n patients—n lesions**
GE Discovery 690	15/38—135/324
Siemens biograph horizon	14/38—133/324
GE Discovery ST	4/38—34/324
Philips Gemini GXL 16	4/38—18/324
GE Discovery 600	1/38—4/324

**Table 2 cancers-14-00984-t002:** The values of HISTO_Skewness, HISTO_Kurtosis, and SUV_max_ (median ± DS, range) for responder and non-responder patients in the three main districts affected by the disease.

District	Responders	Non-Responders	*p*
**Lymph nodes (n = 91)**			
HISTO_Skewness	2.01 ± 2.12 (−1.10–7.66)	3.02 ± 1.44 (0.02–5.60)	**0.006**
HISTO_Kurtosis	11.03 ± 11.79 (1.66–60.40)	13.72 ± 8.85 (1.85–36.05)	**0.028**
SUV_max_	18.67 ± 12.14 (2.88–51.88)	18.16 ± 13.86 (2.77–75.17)	0.738
**Liver (n = 169)**			
HISTO_Skewness	1.35 ± 2.25 (−4.47–7.66)	3.63 ± 1.90 (−0.51–7.63)	**0.0001**
HISTO_Kurtosis	9.04 ± 11.90 (1.81–60.40)	19.34 ± 13.86 (1.75–60.09)	**0.0001**
SUV_max_	19.39 ± 10.17 (4.91–55.86)	20.87–10.14 (9.12–55.26)	0.326
**Bone (n = 42)**			
HISTO_Skewness	2.40 ± 1.89 (0.51–6.67)	4.03 ± 1.87 (0.49–7.74)	**0.014**
HISTO_Kurtosis	11.57 ± 12.83 (2.35–48.00)	23.13 ± 15.46 (2.17–61.34)	**0.015**
SUV_max_	10.31 ± 9.41 (2.06–36.07)	28.42 ± 28.61 (1.67–93.50)	**0.047**

**Table 3 cancers-14-00984-t003:** The values of ΔHISTO_Skewness and ΔHISTO_Kurtosis (median ± DS, range) for PRRT responder and non-responder lesions in the three main districts affected by the disease.

District	Responders	Non-Responders	*p*
**Lymph node (*n* = 91)**			
ΔHISTO_Skewness	21.18 ± 265.75% (−880.0–1533.3)	176.83 ± 469.34% (−96.3–2550.0)	0.886
ΔHISTO_Kurtosis	13.97 ± 83.08% (−82.9–340.5)	−4.48 ± 40.84% (−85.2–96.2)	0.604
**Liver (*n* = 169)**			
ΔHISTO_Skewness	−17.72 ± 865.36% (−6300.0–4800.0)	134.23 ± 324.32% (−180.00–1203.82)	**0.031**
ΔHISTO_Kurtosis	9.76 ± 52.45% (−94.83–193.68)	14.64 ± 60.64% (−94.07–175.68)	0.906
**Bone (n = 42)**			
ΔHISTO_Skewness	6.84 ± 70.95% (−125.0–134.53)	−24.54 ± 71.06% (−240.8–56.6)	0.334
ΔHISTO_Kurtosis	66.15 ± 113.10% (−28.1–338.5)	−0.33 ± 41.43% (−55.7–103.7)	**0.022**

## Data Availability

Data are available for bona fide researchers who request it from the authors.

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
