# Peer review of "[68Ga]DOTATOC PET/CT Radiomics to Predict the Response in GEP-NETs Undergoing [177Lu]DOTATOC PRRT: The “Theragnomics” Concept"

_cancers, 2022, doi:10.3390/cancers14040984_

Round 1

Reviewer 1 Report

The manuscript considers an important problem and would be valuable to the readers of Cancers. The manuscript presented for the review is well elaborated on, but there are some remarks.

1) The Authors use term ”innovative laboratory essays”  (line 34) I wonder whether word ”essays” should be replaced by ”assays”. Please correct it, if necessary.

2) In the part of the manuscript text entitled “Radiomics [68Ga]DOTATOC PET/CT Analysis”, (line 169 -173), standardized-size ROIs were described as ‘circular’. I suggest changing their descritption to ‘spherical’ instead, because of their 3D nature (this should also be changed in later parts of the manuscript).

Please inform me whether the Authors considered the use of automatic segmentation of leasions to recieve ROIs instead of placing a standarized size ROIs.

3) (line 180-181) The Authors calculated the point-biserial correlation between assessed features and dychotomic outcome, but some clinical features, such as grading and number of PRRT cycles, are not continuously measured variables. A point-biserial correlation can only be used to assess the association between one continuous variable and one dichotomous variable. Please clarify this.

PBC also assumes that continuous variable is normally distributed and has equal variances for each category of the other variable. Description should state that these assumptions were confirmed with appropriate tests.

4) (line 186-191) What was the value of k in k-fold cross validation strategy? If “leave-one-out” was applied, please clarify it.

5) Figure 2 contains poor quality image with noise and JPEG compression artifacts. Please replace it with a better quality image, preferably in lossless format (like PNG). Also, replace the PET image (1) with image containing visible ROIs placed on the lesion.

6) How many lesions in G1, G2 and G3 patients, respectively, responded to the therapy?

Author Response

We sincerely thank the editor and the reviewers for their suggestions and time dedicated to our manuscript.

Reviewer 1: The manuscript considers an important problem and would be valuable to the readers of Cancers. The manuscript presented for the review is well elaborated on, but there are some remarks.

R1.1) The Authors use term ”innovative laboratory essays”  (line 34) I wonder whether word ”essays” should be replaced by ”assays”. Please correct it, if necessary.

A1.1) Sorry for this typo. We corrected the whole manuscript accordingly.

R1.2) In the part of the manuscript text entitled “Radiomics [68Ga]DOTATOC PET/CT Analysis”, (line 169 -173), standardized-size ROIs were described as ‘circular’. I suggest changing their descritption to ‘spherical’ instead, because of their 3D nature (this should also be changed in later parts of the manuscript).

A2.2) We apologize because we were not clear in the manuscript. In this study, also due to the heterogeneity, and small dimensions of GEP NET lesions we placed a 2D-circular ROI (minimum size of at least 0.443 cm3, 16 voxels) on the most active area of each lesion. We improved the manuscript accordingly.

R1.3) Please inform me whether the Authors considered the use of automatic segmentation of leasions to recieve ROIs instead of placing a standarized size ROIs.

A1.3) Thank you for this point. In this study, we used a standardized ROI positioning strategy as explained before. However, in future studies we will use deep learning algorithms to automatically segment lesions. In this regard, we added text in the Discussion Section as follow:

The use of deep learning algorithms might also allow us to eliminate any potential time -consuming ROI placement. The deep learning algorithm will be responsible for the entire radiomics process in a completely automatic way, from the segmentation process to the feature extraction process to the pre-dictive model implementation, avoiding the use of LifeX or similar software, and, consequently, also eliminating any user-dependence.”

R1.4) (line 180-181) The Authors calculated the point-biserial correlation between assessed features and dychotomic outcome, but some clinical features, such as grading and number of PRRT cycles, are not continuously measured variables. A point-biserial correlation can only be used to assess the association between one continuous variable and one dichotomous variable. Please clarify this. PBC also assumes that continuous variable is normally distributed and has equal variances for each category of the other variable. Description should state that these assumptions were confirmed with appropriate tests.

A1.4) Thank you for this observation. We apologize because regarding this, we only cited the conference paper in which we proposed this new approach but still in a preliminary step [Comelli A, Stefano A, Coronnello C et al (2020) Radiomics: A New Biomedical Workflow to Create a Predictive Model. Springer International Publishing, Cham, pp 280-293]. The pbc has been adopted as described in Barone, et al. ["Hybrid descriptive-inferential method for key feature selection in prostate cancer radiomics", https://doi.org/10.1002/asmb.2642]: in this work, a novel method for features selection in radiomics was proposed as the combination of descriptive and inferential statistics. Its validity is illustrated through a study on prostate cancer analysis. This paper shows as the descriptive statistic model designed for continuous variables can be used with categorical variables (as reported in the above-mentioned paper “The extracted features obtained by the Mazda software are continuous and categorical variables”) and no assumptions on normally distributed and equal variances of the continuous variable are necessary. We apologize again for the oversight. We have added the reference accordingly.

R1.5) (line 186-191) What was the value of k in k-fold cross validation strategy? If “leave-one-out” was applied, please clarify it.

A1.5) We apologize, this indeed was not clear in the manuscript. The “leave-one-out” method randomly selects k observations to hold out for the validation set. Using this cross-validation method within a loop does not guarantee disjointed validation sets. To guarantee disjointed validation sets, we used the 'k-fold' approach. Therefore, the dataset was divided into approximately k = 5 subsets, and the holdout method was repeated 5 times. In other words, this method uses k - 1 (e.g. 4 for 5 subsets) folds for training and the last fold for validation. The method repeats this process k time, each time leaving one different fold for validation. We edited the paper accordingly.

R1.6) Figure 2 contains poor quality image with noise and JPEG compression artifacts. Please replace it with a better quality image, preferably in lossless format (like PNG). Also, replace the PET image (1) with image containing visible ROIs placed on the lesion.

A1.6) Thank you for this suggestion. We improved figure 2 accordingly.

R1.7) How many lesions in G1, G2 and G3 patients, respectively, responded to the therapy?

A1.7) Thank you for this suggestion, we added the number of responsive lesions to PRRT (SD, PR, CR) according to grading in table 1.

Reviewer 2 Report

cancers-1583293

[68Ga]DOTATOC PET/CT Radiomics to Predict the Response in 2 GEP-NETs Undergoing [177Lu]DOTATOC PRRT: The “Theragnomics” Concept

The authors present a method for predicting the response of neuroendocrine tumours to radionuclide therapy using radiomic features extracted from PET/CT image data. The article would be of interest to the readership of cancers, but would benefit from edits prior to publication in its current form.

[1] How were the 65 features that were extracted from the PET/CT data selected?

[2] Wouldn’t considering each patient’s lesion as a singular add bias to your results?

[3] Could the authors include text or a figure as to how this sort of method could be used prospectively in the current clinical workflow? What kind of clinical decisions would be possible if a prediction of tumour response would be had?

[4] How long does feature extraction take? Would this type of model fit into the current clinical workflow?

[5] One or two spelling mistakes (“assay” spelled like “essay”, etc).

Author Response

We sincerely thank the editor and the reviewers for their suggestions and time dedicated to our manuscript.

Reviewer 2: The authors present a method for predicting the response of neuroendocrine tumours to radionuclide therapy using radiomic features extracted from PET/CT image data. The article would be of interest to the readership of cancers, but would benefit from edits prior to publication in its current form.

R2.1) How were the 65 features that were extracted from the PET/CT data selected?

A2.1) Thank you for this suggestion. LifeX is an analysis software compliant with the Image Biomarker Standardization Initiative (IBSI) that allows the automatic extraction of radiomics features from biomedical images. We imported the PET images, drawing the ROIs around the most active part of the lesion target. Then, as described in Section 2.5, an absolute intensity rescaling factor of 0–60 of the SUV of the ROI was applied, with 64 bins fixed. In this way, 65 radiomics features were automatically obtained for each lesion. The extracted features’ complete list is provided in table S1. Successively, the mixed descriptive-inferential sequential approach described in [26] was used for the feature selection and reduction process. These informations have been added in Section 2.5 as follow:

LifeX [25] is an analysis software compliant with the Image Biomarker Standardization Initiative (IBSI) [Zwanenburg, A.; Vallières, M.; Abdalah, M.A.; Aerts, H.J.W.L.; Andrearczyk, V.; Apte, A.; Ashrafinia, S.; Bakas, S.; Beukinga, R.J.; Boellaard, R.; et al. The image biomarker standardization initiative: Standardized quantitative radiomics for high-throughput image-based phenotyping. Radiology 2020, 295, 328–338] that allows the automatic extraction of radiomics features from biomedical images. For each patient, all [68Ga]DOTATOC-positive lesions that were clearly discriminated, non-confluent, and of minimal size of 16 voxels were selected.  PET images were imported to LifeX and a 2D-circular regions of interest (ROIs) were drawn around every lesion. ROIs had a minimum size of 0.443 cm3(corresponding to at least 16 voxels) to allow for a consistent textural feature calculation. ROI size was adjusted to the size of the lesions, without incorporating adjacent tissue. In this way, using an absolute intensity rescaling factor of 0–60 of the SUV (64 bins, 0.95 fixed bin width) 65 radiomics features were automatically extracted for each lesion.

R2.2) Wouldn’t considering each patient’s lesion as a singular add bias to your results?

R2.3) Could the authors include text or a figure as to how this sort of method could be used prospectively in the current clinical workflow? What kind of clinical decisions would be possible if a prediction of tumour response would be had?

A2.2-2.3) Thank you for these interesting points. To achieve a clinical benefit, the identification of new biomarkers to assess PRRT efficacy and avoid patient toxicity is crucial, but also challenging in a heterogeneous group of tumors such as NENs. Tumor heterogeneity refers to the coexistence of cellular populations bearing different genetic or epigenetic alterations within the same lesion, or in different lesions of the same patient. Intratumor heterogeneity is characterized by its dynamic changes. Therefore, a single biopsy is unlikely to capture the complete genomic landscape of a patient’s tumor considering the spatial-temporal changes in tumor heterogeneity. Similarly, the assessment of the whole patient response instead of the single lesion. Functional imaging and new approaches in image analysis, such as radiomics, could play a key role as prognostic biomarkers and for the therapy response assessment, helping to evaluate intratumor heterogeneity (ITH), which is a spatial and temporal phenomenon more or less distinct in every single lesion. The detection of ITH is the most complex issue to assess PRRT efficacy, and in our study, the assessment of each lesion is aimed at identifying those parameters that can detect earlier and more accurately single PRRT-unresponsive lesions that may benefit from the use of combined/other treatments. As we already stated in the text “The opportunity to assess for each patient the single lesion’s heterogeneity and predict each lesion’s response to PRRT would enhance physicians to early address patients to the best options of care, reducing costs and potential toxicities [11], improving quality of life and survival”. In other words, our per-lesion analysis should be considered as an end-point rather than a bias. However, in future studies we will also assess also per-patient analysis in larger cohorts. The individual assessment (inter-patient heterogeneity, not limited to the single lesion) of these parameters needs future validation with a larger study sample. Therefore, our results if confirmed and externally validated in extensive cohorts may justify the implementation of such radiomics parameters as a tool for the GEP-NET assessment and therapeutical decisions, within tools such as decision nomograms.

R2.4) How long does feature extraction take? Would this type of model fit into the current clinical workflow?

A2.4) The whole process duration (segmentation and features extraction) depends on the number of lesions and on the physician’s expertise/experience. However, once validated this process may be also automatized thus reducing any potential time-consuming and inter-reader’s limitations. It is possible to program the process by automatically placing an ROI over the most active area of each lesion and, consequently, extracting the radiomics features via script. Namely, our next step will be to implement a deep-learning-based algorithm to implement the predictive model without the use of LifeX (or other similar software) to automate the whole radiomics process eliminating any potential time consuming and inter-reader’s constraints. This observation has been added in the Discussion Section as follow:

The use of deep learning algorithms might also allow us to eliminate any potential time -consuming ROI placement. The deep learning algorithm will be responsible for the entire radiomics process in a completely automatic way, from the segmentation process to the feature extraction process to the pre-dictive model implementation, avoiding the use of LifeX or similar software, and consequently, also eliminating any user-dependence.”

R2.5) One or two spelling mistakes (“assay” spelled like “essay”, etc).

A2.5) Sorry for this typo. We corrected this and other typos in the whole manuscript accordingly.